# An evaluation of Chile's Law of Food Labeling and Advertising on sugar-sweetened beverage purchases from 2015 to 2017: A before-and-after study

**Lindsey Smith Taillie**[1,2], **Marcela Reyes**[3], **M. Arantxa Colchero**[4], **Barry Popkin**[1,2], **Camila Corvalán**[3]*

**1** Carolina Population Center, University of North Carolina at Chapel Hill, Chapel Hill, North Carolina, United States of America, **2** Department of Nutrition, Gillings School of Global Public Health, University of North Carolina at Chapel Hill, Chapel Hill, North Carolina, United States of America, **3** Institute of Nutrition and Food Technology, University of Chile, Santiago, Chile, **4** Instituto Nacional de Salud Pública, Cuernavaca, Morelos, Mexico

* ccorvalan@inta.uchile.cl

**Data Availability Statement:** Data are from Kantar WorldPanel Chile (http://www.kantarworldpanel.com/cl). The authors are not legally permitted to

## Abstract

### Background

Chile's Law of Food Labeling and Advertising, implemented in 2016, was the first national regulation to jointly mandate front-of-package warning labels, restrict child-directed marketing, and ban sales in schools of all foods and beverages containing added sugars, sodium, or saturated fats that exceed set nutrient or calorie thresholds. The objective of this study is to evaluate the impact of this package of policies on household beverage purchases.

### Method and findings

In this observational study, monthly longitudinal data on packaged beverage purchases were collected from urban-dwelling households (*n* = 2,383) participating in the Kantar Word-Panel Chile Survey from January 1, 2015, to December 31, 2017. Beverage purchases were linked to nutritional information at the product level, reviewed by a team of nutritionists, and categorized as "high-in" or "not high-in" according to whether they contained high levels of nutrients of concern (i.e., sugars, sodium, saturated fat, or energy) according to Chilean nutrient thresholds and were thus subject to the law's warning label, marketing restriction, and school sales ban policies. The majority of high-in beverages were categorized as such because of high sugar content. We used fixed-effects models to compare the observed volume as well as calorie and sugar content of postregulation beverage purchases to a counterfactual based on preregulation trends, overall and by household-head educational attainment. Of households included in the study, 37% of household heads had low education (less than high school), 40% had medium education (graduated high school), and 23% had high education (graduated college), with the sample becoming more educated over the study period. Compared to the counterfactual, the volume of high-in beverage purchases decreased 22.8 mL/capita/day, postregulation (95% confidence interval [CI] −22.9 to −22.7;

share the data used for this study, but interested parties may contact Kantar WorldPanel representative Maria Paz Roman to inquire about accessing this proprietary data (mariapaz. roman@kantarworldpanel.com). No accession number is needed when requesting data.

**Funding:** Funding support comes from Bloomberg Philanthropies (https://www.bloomberg.org/; received by BP) and the International Development Research Center (Grants 108180 and 107731; https://www.idrc.ca/; received by CC).This research also received support from the Population Research Infrastructure Program awarded to the Carolina Population Center (P2C HD050924) at The University of North Carolina at Chapel Hill by the Eunice Kennedy Shriver National Institute of Child Health and Human Development. The funders had no role in study design, data collection and analysis, decision to publish, or preparation of the manuscript.

**Competing interests:** We have read and understood PLOS Medicine's policy on declaration of interests and LST, MR, CC, and AC declare that they have no competing interests. BP is on the editorial board and otherwise has no competing interests.

**Abbreviations:** CI, confidence interval; FOP, front-of-package; GDA, Guideline Daily Amount; IRB, Institutional Review Board; NFP, nutrition facts panel; SES, socioeconomic status; SSB, sugar-sweetened beverage.

$p < 0.001$), or 23.7% (95% CI −23.8% to −23.7%). High-educated and low-educated households showed similar absolute reductions in high-in beverage purchases (approximately 27 mL/capita/day; $p < 0.001$), but for high-educated households this amounted to a larger relative decline (−28.7%, 95% CI −28.8% to −28.6%) compared to low-educated households (−21.5%, 95% CI −21.6% to −21.4%), likely because of the high-educated households' lower level of high-in beverage purchases in the preregulation period. Calories from high-in beverage purchases decreased 11.9 kcal/capita/day (95% CI −12.0 to −11.9; $p < 0.001$) or 27.5% (95% CI −27.6% to −27.5%). Calories purchased from beverages classified as "not high-in" increased 5.7 kcal/capita/day (95% CI 5.7–5.7; $p < 0.001$), or 10.8% (10.8%–10.8%). Calories from total beverage purchases decreased 7.4 kcal/capita/day (95% CI −7.4 to −7.3; $p < 0.001$), or 7.5% (95% CI −7.6% to −7.5%). A key limitation of this study is the inability to assess causality because of its observational nature. We also cannot determine whether observed changes in purchases are due to reformulation or consumer behavioral change, nor can we parse out the effects of the labeling, marketing, and school sales ban policies.

## Conclusions

Purchases of high-in beverages significantly declined following implementation of Chile's Law of Food Labeling and Advertising; these reductions were larger than those observed from single, standalone policies, including sugar-sweetened-beverage taxes previously implemented in Latin America. Future research should evaluate the effects of Chile's policies on purchases of high-in foods, dietary intake, and long-term purchasing changes.

## Author summary

### Why was this study done?

- In 2016, Chile implemented the Law of Food Labeling and Advertising, a set of policies designed to prevent further increases in obesity prevalence by subjecting foods and beverages high in energy, sugar, sodium, and saturated fat content to marketing restrictions, banned sales in schools, and the first national mandatory front-of-package (FOP) warning-label system.

- Many countries are actively considering implementing similar policies, particularly the FOP warning-label policy.

- Understanding how beverage purchases changed following implementation of this policy package can inform development of future obesity prevention policies.

### What did researchers do and find?

- Using national data on household food purchases from before and after policy implementation, we examined changes in purchases of beverages high in sugar, saturated fat, sodium, or calories (i.e., "high-in" beverages). We compared observed beverage

purchases after policy implementation to expected purchases had the policy not been implemented, based on preregulation trends.

- We found that the purchase volume of high-in beverages decreased by 22.8 mL per capita per day or 23.7% after the regulation was implemented.

-  We also found that although high-educated households and low-educated households had similar absolute reductions in high-in beverage purchases, high-educated households had larger relative reductions in high-in beverage purchases.

### What do these findings mean?

- After Chile's labeling, marketing, and school food sales policies were implemented, purchases of high-in beverages decreased. This observed decrease is greater than purchase changes that have been observed following implementation of single, standalone policies in Latin America, such as a sugar-sweetened-beverage tax.

- Future research will be needed to understand to what degree these changes are attributable to product reformulation of products and/or to changes in consumer behavior, as well as the impact of these regulations on dietary intake and health-related outcomes.

## Introduction

In recent decades, consumption of sugar-sweetened beverages (SSBs) has rapidly increased across the globe [1,2]. Excess intake of these beverages has been linked to increased weight gain, glucose dysregulation, and development of noncommunicable diseases such as type 2 diabetes [3–7]. Compared to foods, SSBs are uniquely harmful because they contain large amounts of calories, which can be rapidly absorbed and are less satiating, leading to inadequate caloric compensation at other eating occasions and contributing to overall positive energy balance [8]. Around the world, public policies are an important and increasingly common strategy being used to reduce consumption of these beverages and prevent continued increases in obesity and related diseases [9–12]. Recently, fiscal policies such as taxes on SSBs have been the predominant approach for reducing intake [11]. Over 42 countries and six United States cities have implemented SSB taxes or increased an existing SSB tax, with most of these implemented in the last decade [13]. Although impact on SSB purchases depends on tax design and rates, these policies have generally led to reductions in SSB purchases equivalent to the price increase on SSBs [14–18]. Evidence also suggests that SSB taxes affect low-income groups and high-SSB consumers the most [16,17,19].

 Additional policy strategies to reduce SSB consumption include mandatory front-of-package (FOP) warning labels, governmental restrictions on marketing SSBs, and bans on SSB sales and promotion in schools [20]. In particular, FOP warning labels have become the focus of many health scholars and advocates, gaining favor over the heretofore more common positive FOP labels (e.g., a health seal or stamp) and other voluntary FOP systems such as traffic-light labels [21–26]. Despite global momentum behind FOP warning-label policies, virtually no evidence exists yet on changes in SSB purchases following implementation of a warning-label policy [27]. Similarly, little is known about the effects of national school sales bans or mandatory

restrictions on unhealthy food marketing to children and subsequent changes in food and beverage purchases [28]. Finally, despite calls for implementation of more comprehensive "packages" of obesity prevention policies [10], the joint impact of a set of policies on SSB purchases is unclear, as most national-level food and beverage policies been implemented one at a time.

Chile, a high-income country with high levels of SSB intake [1,29], has implemented a uniquely comprehensive set of obesity prevention policies regulating how SSBs and other energy-dense, nonessential foods are packaged, marketed, and sold. The first of these regulations, implemented October 2014, increased Chile's existing tax on SSBs from 13% to 18% for high-sugar beverages and decreased the tax from 13% to 10% for low-sugar beverages. One evaluation found that in the first year postimplementation, this modification resulted in only small price increases and 3.4% declines in SSB purchases [30]. In June 2016, Chile implemented the Law of Food Labeling and Advertising, which included the first national system of mandatory FOP warning labels for SSBs and energy-dense, nonessential foods [31]. Similar warning-label policies have since been adopted in Peru, Uruguay, and Israel in 2018, and policies are currently under public review/potential finalization phases in Brazil and Mexico, among others. Chile's Law of Food Labeling and Advertising also exacts comprehensive restrictions on child-directed marketing of SSBs and nonessential, energy-dense foods to children under 14 years of age, as well as restrictions on the promotion and sales of these products in schools [31]. Chile's marketing regulation, in particular, is more comprehensive than other countries' in that it restricts unhealthy food marketing on more products across a wider range of media, and it prohibits use of more marketing techniques [28]. Understanding how Chile's policies on labeling and marketing—and, to a lesser extent, the school sales ban—are linked to changes in household purchases is critical for developing evidence-based obesity prevention policies across the globe.

It is also important to understand whether these policies had a differential effect by socioeconomic status (SES) to ensure that such policies do not inadvertently increase SES-related diet disparities, especially in regions such as Latin America, where the burden of obesity is shifting to lower-SES individuals and households [32–34]. Education is a particularly important measure of SES when evaluating labeling policies, as education may influence purchasing decisions by affecting how well an individual is able to understand the information communicated in nutrition education messages or on food labels [35–37]. Education is also correlated with higher financial and nonfinancial resources [38], which can likewise influence food choices. Moreover, it is currently unclear whether low- versus high-educated households would have differential responses to other policies, such as marketing restrictions or school sales bans on high-in beverages. Thus, it is important to examine whether low- versus high-educated households show greater changes in beverage purchases after Chile's labeling regulation was implemented.

This study's objective is to use longitudinal data on household beverage purchases made in stores to examine changes in the volume, calorie content, and sugar content of beverage purchases following implementation of the Chilean labeling and marketing regulation, overall and by household educational attainment.

## Methods

This study was reviewed and approved by the University of Chile Institutional Review Board (IRB) and is exempt from review by the University of North Carolina, Chapel Hill IRB as the study uses secondary, de-identified data. This study and protocol were registered with the Open Science [39] and protocols.io [40].

## Participants

This study uses data on household beverage purchases from January 1, 2015, to December 31, 2017, from the Kantar WorldPanel Chile. The response rate for participation in Kantar World-Panel Chile is 95%. Households are excluded from participation if any member is engaged in activities determined to have potential to interfere with the collection of information about the products studied (e.g., a household member works at an advertising agency, market research company, or written or spoken media company or is owner of a company that markets a product studied). Households are also excluded if they do not meet minimum purchasing standards (e.g., purchase at least one of item from one of the 15 categories from the "basic basket" of goods).

Data include household purchases of consumer packaged goods from a panel of 2,000 households located in cities with >20,000 inhabitants, representative of Chile's urban population [41]. With replacement, our analytic sample had 2,383 unique households, with an average follow-up of 29.2 months, providing 69,696 household-month observations. Enumerators visited households weekly to collect data on food and beverage purchases. Information on each purchase was collected either by scanning product barcodes using a handheld barcode scanner or by using a codebook to assign barcodes for bulk products or other products without barcodes. Interviewers also reviewed weekly receipts, conducted household pantry inventories, and checked empty product packages stored in a bin between interviews to ensure products were not double-counted. Data collected on each purchase included volume or weight, bar code, price per unit, retail channel, brand, package size, and date of purchase. Data were analyzed at the household-monthly level.

## The Chilean regulation

Details of the Chilean Law of Food Labeling and Advertising have been published previously [31]. This regulation was designed to be implemented in three phases with increasingly stringent nutrient thresholds (S1 Table). The first phase of implementation began in late June 2016. Products subject to the regulation were required to carry FOP warning labels, faced child-directed marketing restrictions, and were banned from sales and promotion in schools and nurseries. Briefly, the FOP warning labels consist of a black octagon with white borders placed on the front of the food or beverage package, including the words "alto en. . ." ("high in. . .") calories, sugars, saturated fats, or sodium (S1 Fig). The marketing restriction includes a ban on the use of child-directed marketing techniques in any communication channel and the advertisement of these products on children's television programs and websites (i.e., programs and websites with >20% of the audience <14 years of age). The school restriction prohibits sales of high-in products on school grounds; high-in products also cannot be offered as part of the school feeding program.

## Nutrition facts panel data and categorization by regulation status

A visual depiction of the Chilean regulation timeline overlaid with the timing of study data collection can be seen in S2 Fig.

We obtained nutrition facts panel (NFP) data from product photographs collected by a team of Chilean nutrition research assistants in stores during the first quarters of 2015, 2016, and 2017 [42]. We then linked NFP data at the product level to household beverage purchases using a similar process as described in previous household purchase evaluations [43,44]. For the preregulation period, we linked purchases to NFP data collected in 2015 and 2016 (i.e., data reflecting the nutritional profiles of products available prior to the regulation). For the postregulation period, we linked purchases to NFP data collected in 2017. If there was no

direct 2017 link, we linked the product to the 2015–2016 NFP data. Linkages were based on barcode, brand name, and product description. Of total beverage purchases, 95.6% were linked to collected NFP data. If no collected NFP data were available for a purchased product, it was linked to Mintel Latin America (4.4%) or other NFP data resources (<0.1%).

After linking the data, a team of Spanish-speaking nutritionist research assistants at the University of North Carolina and the University of Chile categorized each beverage purchase as to whether it should be subject to regulation according to the first-phase nutrient profile model established by the Chilean regulation (S1 Table). Beverages were categorized as "high-in" (and thus subject to regulation) if they contained added sugar, added sodium, or added saturated fat and exceeded the nutrient thresholds set in the first phase of implementation (i.e., >100 calories, >100 mg sodium, >6 g sugar, or 3 g saturated fat per 100 mL of product in its as-consumed form). These "high-in" beverages would be required to carry an FOP warning label and be subject to the regulation's marketing and school sales restrictions. Beverages were considered "not high-in" if they did not meet these nutritional criteria. Most beverages that were categorized as high-in were classified as such because they contained added sugar and their total sugar content exceeded 6 g/100 mL. We also classified beverages into subgroups, including sodas, fruit drinks, dairy products, waters, coffees and teas, 100% fruit and vegetable juices, and sports and energy drinks (S2 Table).

Beverage purchases were considered to have occurred in the preregulation period if they were purchased between January 2015 and June of 2016 and in the postregulation period if they were purchased between July 2016 and December 2017.

### Outcomes

The main outcomes were average per capita daily volume (mL), calories (kcal), and sugars (g) purchased from high-in, not-high-in, and total beverages. As described below, for each outcome we compared the observed adjusted mean postregulation purchase amount to a counterfactual, or what we estimate would have been observed in the postregulation period based on preregulation trends predicted into the postregulation period. This approach is similar to that used in previous SSB policy evaluations [16,30].

### Covariates

Main covariates included household education level (self-reported by head of household and categorized as low [less than high school], middle [completed high school], or high [completed college or higher]); age of head of the household; an assets index (created using factor analysis based on number of household rooms, bathrooms, and vehicles owned—specified as a continuous variable for main analyses, and a 3-level categorical variable based for sensitivity analyses on household assets); household composition (specified as a set of discrete variables, each with the number of people in the following age categories: children 0–1 years, children 2–5 years, children 6–13 years, males 14–18 years, females 14–18 years, female adults over 18 years, and male adults over 18 years); and month dummies to adjust for seasonality. Because trends in economic activity could influence beverage purchases, we also controlled for monthly, region-level unemployment rates [45].

### Statistical analyses

All analyses were conducted using Stata 14 (College Station, TX, USA). The prespecified analysis plan was published on November 6, 2018, in the Open Science Framework (https://osf.io/fuh63/).

## Unadjusted analyses: Descriptive statistics

First, we examined sociodemographic characteristics of the households participating in Kantar WorldPanel Chile each year. We examined unadjusted proportions of households that purchased high-in and not-high-in beverages in a given month (i.e., purchases > 0 mL), before and after implementation of the regulations. We also examined pre- and postregulation mean unadjusted volume, calories, and sugar purchased from high-in and not-high-in beverage purchases, overall and by beverage subgroup.

## Adjusted analyses: Fixed-effects models

Because the Chilean regulations were implemented nationally, all members of the population were exposed to the policy at the same time, precluding a randomized controlled experimental design. Thus, we used a pre-post quasi-experimental modeling approach to examine changes in the average household beverage purchases that occurred before and after policy implementation. Similar to previous evaluations [16,30,46,47], and following the methods of interrupted time series analyses [48], we constructed a counterfactual by including in the model an interaction between a count variable for time and a binary variable for regulation period (pre versus post). This counterfactual represents the average predicted household beverage purchases in the postregulation period based on preregulation trends projected into the postregulation period (i.e., what was expected without a regulation implemented in 2016).

To determine the time period to be used for the pre-post comparisons, we ran models using 10- to 18-month windows before and after the regulation, with different ways to account for time and seasonality (i.e., month as an indicator variable or continuous variable, quarterly indicator variables for seasonality). Models with less than 13 months were sensitive to how seasonality was specified, although all models had similar results when using month as a continuous variable. Because the 18-month window was less sensitive to seasonal specifications and allowed for a longer preregulation period in which to estimate postregulation trends, we selected the 18-month window for our main model (S3 Table). Thus, in our final model, the pre-period was specified as January 1, 2015, to June 30, 2016, and the postregulation period was specified as July 1, 2016, to December 31, 2017.

For each outcome (volume, calories, and sugar per capita per day), we calculated the average absolute and relative differences between the observed trend and the counterfactual in the postregulation period based on predicted values from the model. For each primary outcome, we report 95% confidence intervals (CIs) [49]. For main results, we also calculated p-values based on t tests between the observed absolute values and the predicted absolute values from each model in the postregulation period. We used fixed-effects models to account for non-time-varying unobserved household characteristics (e.g., preferences for beverages) and controlled for the aforementioned time-varying household characteristics and contextual covariates. For high-in beverages that had >10% of nonpurchases in a given month, we used two-part models to account for the higher frequency of zero purchases [50]. Two-part models were not necessary for not-high-in and total beverages because these beverages were purchased by more than 90% of the households. For all models, we specified robust standard errors to account for intrahousehold correlation (i.e., to account for repeated measures of households over time).

Because of the skewed distribution of beverage purchases, we used the logarithm of beverage purchases as outcomes for models. Then, to allow for interpretability, we back-transformed logged outcomes into milliliters (mL), calories (kcal), or grams (g) by applying Duan smearing factors that account for the nonnormal distribution of the error term [51].

Finally, we conducted stratified analyses by household education level. To test for differences in education on changes in purchases of high-in beverages after the regulation was implemented, we ran a fully interacted model including interactions of all variables in the model with education. We then used an F-test to determine whether the triple interaction of education (specified as dummies), the regulation time period (a dummy variable for pre/post), and time (continuous) was statistically significant (e.g., the interaction of postregulation × month/year × education level).

## Sensitivity analyses

We first conducted sensitivity analyses to determine whether results differed using different model specifications (e.g., not logged) or modeling techniques (e.g., generalized estimating equation with a log link).

Second, we conducted sensitivity analyses which included price as a covariate in the models. Because the objective of this paper was to understand changes in household beverage purchases before and after implementation of the law, regardless of mechanism (i.e., changes in industry behavior or consumer behavior), we did not include price changes in our main models. However, considering that prices may have changed over time, we conducted a sensitivity analysis controlling for changes in prices to understand whether their inclusion altered results. We derived unit values (prices) by dividing household expenditures over quantity purchased. To reduce measurement error at the household level and because unit values are not weighted, we aggregated them at the month/year/region level and included this variable in the model.

Third, in the main analyses, household beverage purchases made after the policy implementation (July 1, 2016) were preferentially linked to NFP data collected in the postregulation period (first quarter of 2017). However, 2017 NFP data were collected 6 months after implementation, and it is not clear when postregulation product reformulations might have occurred (i.e., whether reformulations occurred closer to the July 2016 implementation date or closer to NFP data collection in January–March 2017). Thus, we conducted additional analyses in which we linked purchase data from the 6-month period immediately following implementation of the regulation but prior to NFP data collection (i.e., July 1, 2016–December 31, 2016) to the preperiod NFP data, in order to understand differences in purchases if reformulations were not incorporated during this period.

Fourth, Chile modified its soda tax from 13% to 18% for industrialized beverages containing >6.25 g sugar/100 mL and from 13% to 10% for industrialized beverages containing <6.25 g sugar/100 mL in October of 2014. Although our previous evaluation of this tax found only small reductions (3.4%) in purchases of high-taxed beverages in the first year, we conducted a sensitivity analysis in which we shifted the preregulation period from January 1, 2015–June 30, 2016 to January 1, 2014–June 30, 2016 to account for any potential changes in beverage purchases in the preregulation period due to the tax modification. These models also included a tax-period indicator variable (tax period = 0 for January 1, 2014–September 30, 2014; tax period = 1 for October 1, 2014, onwards).

Finally, SES is a complex measure that is often represented by a number of variables, including education, income, assets, or others [52,53]. Whereas our main analyses use education as a proxy for SES, we also conducted stratified models by tertiles of the household assets index to understand whether there were any differences depending on how we measured SES.

**Table 1. Weighted household characteristics in the Kantar WorldPanel Chile analytical sample, 2015 to 2017.**

| Characteristics | 2015 | 2016 | 2017 |
|---|---|---|---|
| No. unique households | 2,099 | 2,077 | 2,100 |
| Household-months of observations | 23,401 | 23,456 | 22,839 |
| Head-of-household education (%) | | | |
| <High school | 36.8 | 32.2 | 30.8 |
| High school | 39.9 | 42.7 | 42.5 |
| College or greater | 23.4 | 25.2 | 26.7 |
| Household assets index[1] (%) | | | |
| Low | 34.7 | 36.2 | 30.8 |
| Middle | 31.9 | 30.2 | 35.2 |
| High | 33.4 | 33.6 | 34.0 |
| Household composition, by sex and age (mean ± SE) | | | |
| Children 0–1 year | 0.1 ± 0.01 | 0.1± 0.01 | 0.0 ± 0.00 |
| Children 2–5 years | 0.4 ± 0.01 | 0.4 ± 0.01 | 0.4 ± 0.01 |
| Children 6–13 years | 0.6 ± 0.02 | 0.6 ± 0.02 | 0.6 ± 0.02 |
| Any child <14 years | 1.1 ± 0.02 | 1.1 ± 0.02 | 1.1 ± 0.02 |
| Males 14–18 years | 0.2 ± 0.01 | 0.2 ± 0.01 | 0.2 ± 0.01 |
| Females 14–18 years | 0.2 ± 0.01 | 0.2 ± 0.01 | 0.2 ± 0.01 |
| Men | 1.2 ± 0.02 | 1.2 ± 0.02 | 1.3 ± 0.02 |
| Women | 1.5 ± 0.02 | 1.5 ± 0.02 | 1.6 ± 0.02 |
| Region (%) | | | |
| Santiago | 47.9 | 47.9 | 48.0 |
| North | 12.7 | 12.7 | 12.7 |
| Valparaiso | 11.9 | 11.9 | 11.9 |
| Central South | 8.6 | 8.6 | 8.6 |
| Bio-Bio | 10.5 | 10.4 | 10.3 |
| South | 8.5 | 8.5 | 8.6 |
| Monthly regional unemployment rate (mean ± SE) | 6.3 ± 0.00 | 6.1 ± 0.00 | 6.5 ± 0.00 |
| Volume beverage purchases (mL/capita/day; mean ± SE) | | | |
| High-in beverages | 127.8 ± 1.9 | 103.7 ± 1.8 | 83.5 ± 1.8 |
| Not-high-in beverages | 291.4 ± 1.0 | 285.2 ± 0.8 | 286.6 ± 0.8 |

[1]Low, middle, and high household assets correspond to categories based on tertiles of the household assets index.

## Results

### Unadjusted results

Table 1 provides sociodemographic characteristics of the sample. From 2015 to 2017, the percent of households with low education (less than high school) decreased, whereas the percent of households with middle education (high school degree) or high education (college degree) increased. The unadjusted purchase volume of high-in, not-high-in, and total beverages was declining prior to and after the regulation (S3 Fig).

Comparing the preregulation period to the postregulation period, the percent of households who purchased high-in beverages in a given month declined 10.5 percentage points (95% CI −11.4 to −9.6; $p < 0.001$), from 92.9% in the preregulation period to 82.6% in the postregulation period (S4 Table). The largest declines in percent consumers were observed for high-in fruit drinks (−42.9 percentage points, 95% CI −44.2 to −41.7; $p < 0.001$) and high-in dairy drinks (−28.8 percentage points, 95% CI −30.2 to −27.4; $p < 0.001$). The unadjusted

mean amount of high-in beverages purchased declined by 35.4 mL/capita/day from the preregulation period to the postregulation period (95% CI −39.8 to −30.9; $p < 0.001$), with the largest declines observed among soda (−14.8 mL/capita/day, 95% CI −18.8 to −10.8; $p < 0.001$) and fruit drinks (−14.5 mL/capita/day, 95% CI −15.6 to −13.4; $p < 0.001$) (S5 Table). The percent consumers of not-high-in beverages increased by 1.0 percentage point (95% CI 0.7–1.4; $p < 0.001$) from 96.6% to 97.6%, with the largest increases occurring among fruit drinks (35.4 percentage points, 95% CI 34.2–36.7; $p < 0.001$). Similarly, the largest increase in purchase volume was observed for not-high-in fruit drinks (15.0 mL/capita/day, 95% CI 14.1–16.0; $p < 0.001$). We were unable to examine unadjusted changes in some beverage subgroups because of very low purchase levels before and after the regulation, including 100% fruit and vegetable juice, sports and energy drinks, and high-in coffees.

## Adjusted results

Coefficients from the main regression models are given in S6 Table.

Compared to the counterfactual, adjusted mean volume of high-in beverage purchases decreased by 22.8 mL/capita/day (95% CI −22.9 to −22.7; $p < 0.001$) in the postregulation period, or 23.7% (95% CI −23.8% to −23.7%) (Fig 1).

Stratified models found that high-educated households and low-educated households showed similar absolute reductions in high-in beverage purchases (approximately 27 mL/capita/day, $p < 0.001$ for both comparisons), although for high-educated households this reflected a larger relative decline (−28.7%, 95% CI −28.8% to −28.6%) than for low-educated households (−21.5%, 95% CI −21.6% to −21.4%). Middle-educated households had the lowest reductions in high-in beverage purchases in both absolute and relative terms. In the fully interacted model with low-educated households as the referent group (S7 Table), the interaction between education, regulation period, and time was statistically significant ($p < 0.001$) for the high-educated households but not for the middle-educated households.

Compared to the counterfactual, the adjusted mean volume of not-high-in beverage purchases increased 14.6 mL/capita/day (95% CI 14.6–14.7; $p < 0.001$) or 4.8% (95% CI 4.8%–4.8%) in the postregulation period (Fig 2). Middle-educated households had the greatest absolute and relative increases in not-high-in beverage purchases, with an increase of 21.1 mL/capita/day (95% CI 20.1–21.2; $p < 0.001$) or 7.2% (95% CI 7.2%–7.3%). High-educated households had the smallest increase in not-high-in beverage purchases, both in absolute (9.0 mL/capita/day, 95% CI 8.9–9.1; $p < 0.001$) and in relative terms (2.5%, 95% CI 2.4%–2.5%). Low-educated households increased not-high-in beverage purchases by 11.6 mL/capita/day (95% CI 11.6–11.7; $p < 0.001$) or 4.2% (95% CI 4.2%–4.2%).

Table 2 provides estimated changes in beverage calories and sugar purchased relative to respective counterfactuals for high-in, not-high-in, and total beverages. Calories purchased from high-in beverages decreased by 11.9 kcal/capita/day (95% CI −12.0 to −11.9; $p < 0.001$) or 27.5% (95% CI −27.6% to −27.5%), and sugar purchased from high-in beverages declined by 2.7 g/capita/day (95% CI −2.7 to −2.7; $p < 0.001$) or 25.1% (95% CI −25.1% to −25.0%). In contrast, calories purchased from not-high-in beverages increased 5.7 kcal/capita/day (95% CI 5.7–5.7; $p < 0.001$) or 10.8% (95% CI 10.8%–10.8%), and sugar purchased from not-high-in beverages increased 0.7 g/capita/day (95% CI 0.7–0.7; $p < 0.001$) or 10.2% (95% CI 10.2%–10.2%). Calories purchased from total beverages declined 7.4 kcal/capita/day (95% CI −7.4 to −7.3) or 7.5% (95% CI −7.6% to −7.5%), and sugar purchased from total beverages declined 1.7 g/capita/day (95% CI −1.7 to −1.6) or 10.0% (95% CI −10.1% to −10.0%).

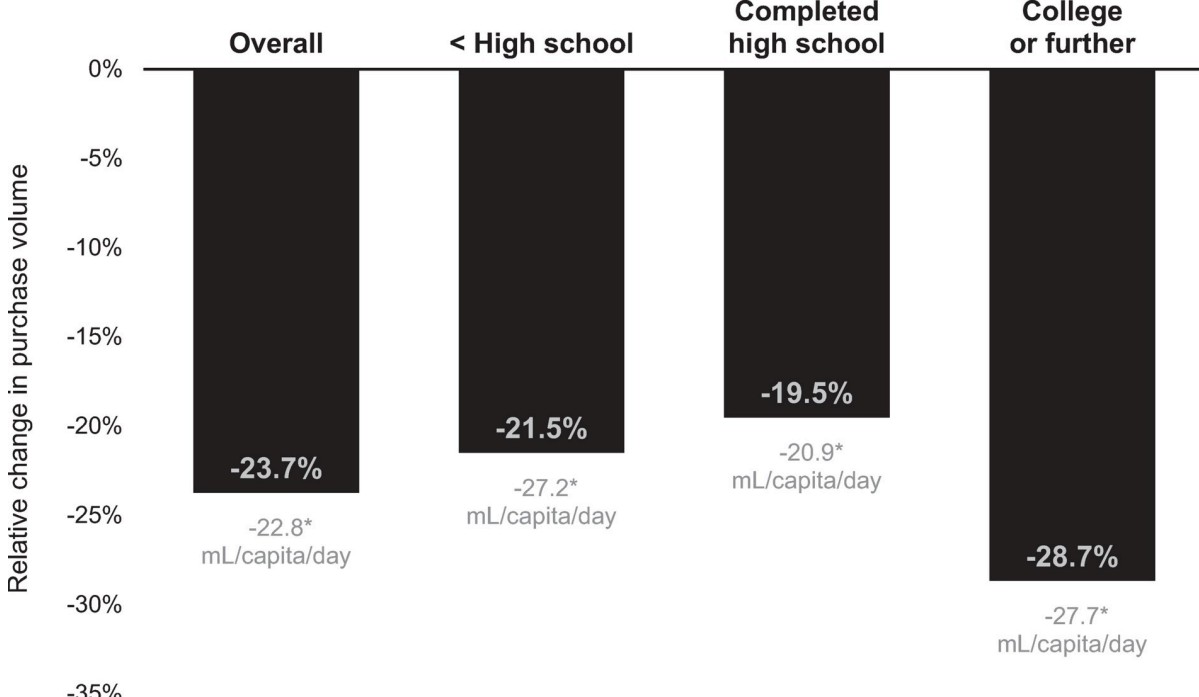

**Fig 1. Relative and absolute changes in purchases of high-in beverages, by education level of household head.** Estimates were derived from fixed-effects models comparing observed postregulation volume of purchases to counterfactual postregulation volume of purchases based on preregulation trends. Purchase data were provided by Kantar WorldPanel Chile. High-in beverages were those subject to the Chilean Law of Food Labeling and Advertising because they contained added sugars, saturated fats, or salt and exceeded nutrient or energy thresholds; not-high-in beverages did not exceed nutrient thresholds and were not subject to the regulation. *$p < 0.001$ for the difference between observed mean absolute values and counterfactual mean absolute values in the postregulation period.

## Comparison to other SSB policies

Fig 3 compares estimated relative and absolute changes in high-in beverage purchases from this evaluation of Chile's Law of Food Labeling and Advertising with findings on the same outcome from previous evaluations of Chile's 5% SSB tax increase [30] and Mexico's 10% SSB tax (1 year [16] and 2 years [17] postimplementation). Relative to their respective counterfactuals, we found larger absolute and relative declines in the volume of high-in beverage purchases under the Chile's Law of Food Labeling and Advertising.

## Sensitivity analyses

S8 Table presents results from two additional model specifications to test the robustness of our findings: fixed-effects models without using Duan smearing factors (exponentiating the predicted values to back-transform logarithm into volume purchased) and generalized estimating equations for panel data with log link function. Both specifications produced greater absolute and relative reductions in high-in beverage purchases compared to our main model. For not-high-in beverages, the specification with no Duan smearing factor showed a larger increase in purchases, but the specification with generalized estimating equations showed a decline in not-high-in beverage purchases, so our main findings are in the range of these two specifications. Our preferred model is fixed effect with Duan smearing factors because it addresses the nonnormal distribution of the error terms, and the results are in the range of these two alternative models while providing more conservative estimates for high-in beverages.

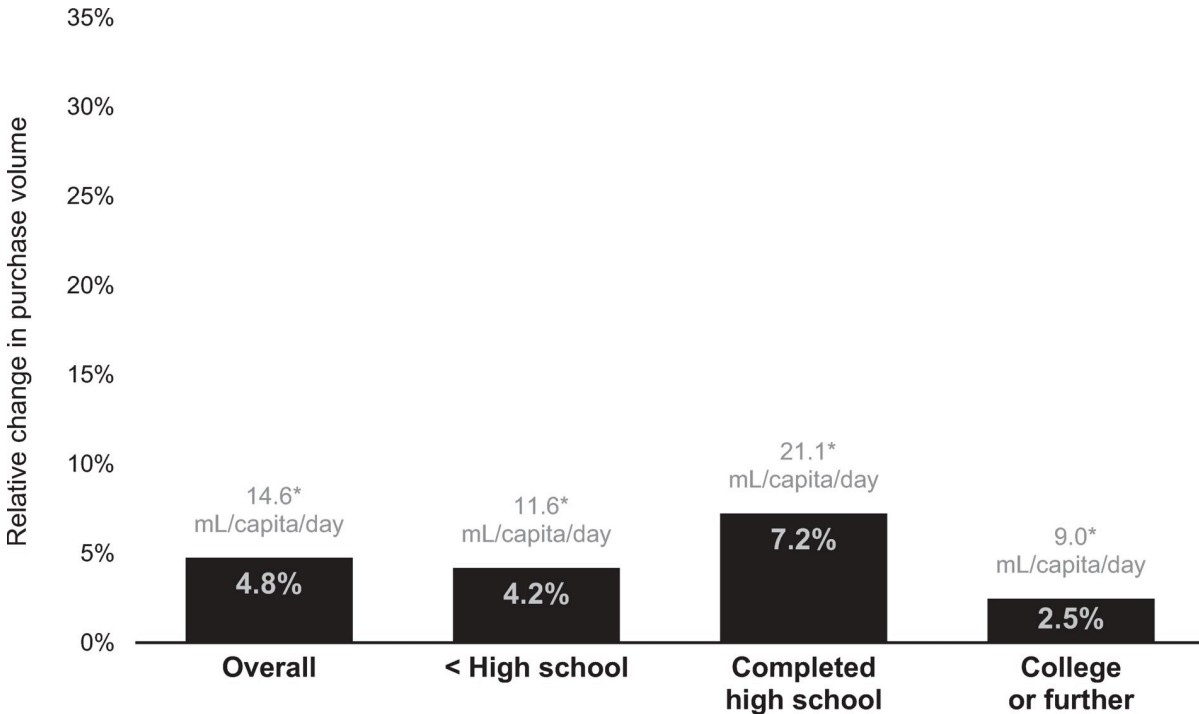

**Fig 2. Relative and absolute changes in purchases of not-high-in beverages, by education level of household head.** Estimates were derived from fixed-effects models comparing observed postregulation volume of purchases to counterfactual postregulation volume of purchases based on preregulation trends. Purchase data provided by Kantar WorldPanel Chile. Not-high-in beverages were not subject to the Chilean Law of Food Labeling and Advertising because they either did not contain added sugars, saturated fats, or salt or they did contain one or more of those added ingredients but did not exceed nutrient or energy thresholds. $^*p < 0.001$ for the difference between observed mean absolute values and counterfactual mean absolute values in the postregulation period.

Sensitivity analyses on the volume of beverages purchased produced results consistent with the main model (S9 Table). Models adjusting for prices found higher postregulation increases in volume of not-high-in beverage purchases, but similar decreases in volume of both high-in and total beverage purchases. Analyses using the preperiod NFP linkage for the 6-month period between

**Table 2. Estimates of average absolute and relative differences in postregulation beverage purchases,[1] comparing observed to counterfactual purchases.**

| | Volume | | Calories | | Sugar | |
|---|---|---|---|---|---|---|
| | Absolute difference | Relative difference | Absolute difference | Relative difference | Absolute difference | Relative difference |
| | mL/capita/day | % | kcal/capita/day | % | g/capita/day | % |
| | (95% CI) | (95% CI) | (95% CI) | (95% CI) | (95% CI) | (95% CI) |
| High-in[2] | −22.8* | −23.7% | −11.9* | −27.5% | −2.7* | −25.1 |
| | (−22.9 to −22.7) | (−23.8 to −23.7) | (−12.0 to −11.9) | (−27.6 to −27.5) | (−2.7 to −2.7) | (−25.1 to −25.0) |
| Not-high-in[3] | 14.6* | 4.8% | 5.7* | 10.8% | 0.7* | 10.2 |
| | (14.6–14.7) | (4.8–4.8) | (5.7–5.7) | (10.8–10.8) | (0.7–0.7) | (10.2–10.2) |
| Total | −8.8* | −2.2% | −7.4* | −7.5% | −1.7* | −10.0 |
| | (−8.8 to −8.8) | (−2.2 to −2.2) | (−7.4 to −7.3) | (−7.6 to −7.5) | (−1.7 to −1.6) | (−10.1 to −10.0) |

[1]Purchase data provided by Kantar WorldPanel Chile.

[2]High-in beverages are those subject to the Chilean Law of Labeling and Advertising because they contain added sugars, saturated fats, or salt and exceed nutrient or energy thresholds.

[3]Not-high-in beverages are not subject to the Chilean Law of Labeling and Advertising because they either do not contain added sugars, saturated fats, or salt or they do contain one or more of those added ingredients but do not exceed nutrient or energy thresholds.

$^*p < 0.001$ for the difference between observed mean values and counterfactual mean values in the postregulation period.

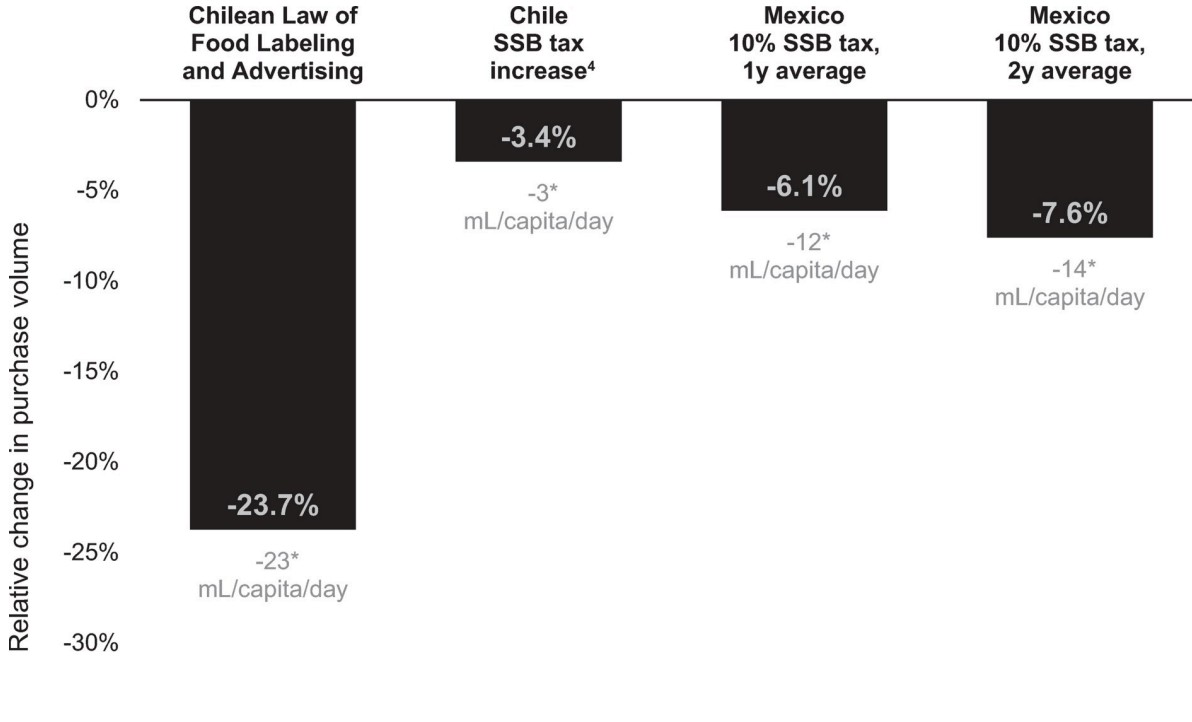

**Fig 3. Relative and absolute changes in purchases of high-in beverages under Chilean and Mexican laws.** Estimates were derived from models comparing observed postregulation volume of purchases to counterfactual postregulation volume of purchases based on preregulation trends. Purchase data provided were by Kantar WorldPanel Chile. High-in beverages were those subject to the Chilean Law of Food Labeling and Advertising because they contained added sugars, saturated fats, or salt and exceeding nutrient or energy thresholds. The Law of Food Labeling and Advertising included mandatory front-of-package warning labels, restrictions on marketing to children, and a ban on sales in schools on all products who met these criteria. [4]Increase from 13% to 18% tax on high-sugar beverages. *$p < 0.001$ for the difference between observed mean absolute values and counterfactual mean absolute values in the postregulation period. SSB, sugar-sweetened beverage.

July 1, 2016, and December 31, 2016, found smaller declines in volume of high-in beverage purchases in both absolute (−16.9 mL/capita/day, 95% CI −17.1 to −16.8; $p < 0.001$) and relative terms (−17.4%, 95% CI −17.5% to −17.3%). Models taking into account the longer preperiod (from 2014) and including a tax dummy for the October 2014 SSB tax modification found larger absolute (−30.0 mL/capita/day, 95% CI −30.1 to −29.9; $p < 0.001$) and relative (−29.2%, 95% CI −29.3% to −29.2%) declines in volume of high-in beverage purchases.

Sensitivity analyses that stratified the models by tertile of household assets index can be found in S4 Fig. Results differed partially from those observed when stratifying by education. Unlike low-education households, those in the lowest tertile of assets index showed a greater decline in absolute volume of high-in beverage purchases (−28.2 mL/capita/day, 95% CI −28.5 to −27.9; $p < 0.001$) compared to households in the top tertile of assets index (−24.5 mL/capita/day, 95% CI −24.7 to −24.3; $p < 0.001$). Like high-education households, those in the highest tertile of assets index showed a greater relative decline (−26.1%, 95% CI −26.2% to −26.0%) than households in the lowest tertile of assets index (−22.2%, 95% CI −22.3% to −22.1%). Households in the middle tertile had the smallest declines in volume of high-in beverage purchases in both absolute and relative terms.

## Discussion

The main finding of this study is that following implementation of the Chilean Law of Food Labeling and Advertising, household purchases of high-in beverages decreased 23.7%

compared to the counterfactual, or what would have been expected based on preregulation trends. This translates into roughly 12 fewer calories and 2.7 fewer grams of sugar purchased per capita per day from high-in beverages. In contrast, volume of not-high-in beverage purchases increased by 4.8%, translating into 5.7 calories and 0.7 grams of sugar purchased per capita per day. This increase in not-high-in beverage purchases was not commensurate with the decrease in high-in beverage purchases. Indeed, there was a relative decline in total beverage purchases of 2.2% for volume, translating into a decrease of 7.4 calories and 1.7 g sugar of beverage purchases per capita per day. Among beverage subcategories, we found the largest reductions in consumption among high-in fruit drinks and dairy (42.9% and 28.8% percentage point reductions in the percent of households consuming these drinks, respectively).

Households with higher educational attainment had larger relative reductions in high-in beverage purchases than did households with less education. This may be because higher-educated household had lower baseline purchases of high-in beverage purchases than did low-educated households, creating a larger relative difference compared to the counterfactual in the postregulation period. However, the interaction term for education, regulation period, and time was also statistically significant for high-educated households versus low-educated households, suggesting higher-education households had a differential response to the regulation. This is consistent with previous literature finding that higher-educated individuals have higher levels of health literacy [54] and may be more likely to use and understand nutrition labels [55–58]. Absolute reductions were similar in amount between highest- and lowest-educated households. These results may be concerning, since similar absolute reductions in high-in beverage purchases could lead to an increase in the relative differences in SSB consumption between high- and low-educated households, since low-educated groups had higher baseline consumption levels. This potential for an increasing disparity in SSB consumption is problematic considering that in Chile, individuals with less education already have higher levels of obesity [59,60]. Future research will be needed to further understand whether these policies increase or reduce disparities in diet and health by educational attainment. Additional research will also be needed to examine changes in purchases among those who were top consumers of high-in beverages prior to the regulation, since previous SSB policy evaluations have found top consumers typically reduce purchases more than do low consumers [47].

Although the Chilean law included a package of policies implemented at once, it is interesting to consider how the effects observed here compare to results from evaluations of previous Latin American policies focused on SSB reduction, primarily SSB taxes. Compared to their respective counterfactuals, SSB purchases in Chile declined by 23.7% under the Chilean Law of Food Labeling and Advertising (22.8 mL/capita/day). This relative decline is larger than or similar to previous estimates of reductions in SSBs after Chile's earlier SSB tax modification, which raised the tax on high-sugar drinks from 13% to 18% (one SSB tax evaluation study found a 3.4% decline in SSB purchases [30], whereas a second study found a 21.6% decline [61], though both studies found only an approximately 2% increase on paid price for SSBs, making the latter study's large drop in SSBs unlikely based on price elasticity [62]). The absolute and relative decline in SSB purchases after the Chilean Law of Food Labeling and Advertising was also larger than the declines in SSB purchases observed after Mexico's 10% SSB tax, which was associated with an average 7.6% posttax reduction in SSB purchases (12 mL/capita/day) 2 years after the tax [17]. The observed decline in the present study was also larger than the average decline in SSB purchases after a 10% SSB tax, as estimated by a recent meta-analysis [63].

On the other hand, it is unclear how these results will compare to larger SSB taxes, which, as expected, are likely to have larger effects on purchases, as indicatd by recent evaluations of Philadelphia's beverage tax [64]. More generally, it is complex to draw comparisons between SSB policies implemented in different settings and times and with different study designs, at

least in part because these real-world evaluations rely on pre-post observational data and cannot assess causality. Based on available evidence, however, the changes in SSB purchases relative to counterfactual estimates following implementation of Chile's Law of Food Labeling and Advertising are larger than those observed after most Latin American SSB taxes. The relatively large effect observed here is not surprising, given that the Chilean Law of Food Labeling and Advertising included a package of policies targeting different aspects of consumer behavior, whereas most SSB taxes have been implemented as standalone policies. These results suggest that policymakers and public health advocates should consider a package of policies, including FOP warning labels, marketing restrictions, and school sales policies alongside SSB taxes as important strategies for reducing population-level purchase and intake of SSBs.

Although there have not yet been any evaluations of a real-world mandatory FOP warning-label policy, the estimated decline in high-in beverage purchases relative to the counterfactual is also larger than what might be expected based on recent meta-analyses of food labeling policies. For example, Shanguan and colleagues found that labeling was only associated with a 6.6% decline in calories purchased [65], whereas Crockett and colleagues largely found a null effect of labeling interventions across outcomes [27]. However, these studies examined a diverse array of labeling systems, including voluntary FOP systems, such as the Guideline Daily Amounts (GDAs) and traffic-light labeling, as well as back-of-package nutritional labeling, in a variety of settings (stores, schools, vending machines, cafeterias, and restaurants). Recent experimental studies in Latin American populations have found that the style of FOP warning labels used in Chile is easier to understand and more likely to discourage consumption than other types of FOP labels, such as the GDAs or traffic-light labels [66–68]. In addition, a recent US-based randomized experiment found that nutrient-based FOP warning labels reduced SSB purchases by 31 calories per interaction, or 22% [69], which is similar to the effect observed in this study.

In addition, along with the warning-label component, the Chilean law includes marketing and school sales restrictions, which likely contributed to the larger effect found here. A recent evaluation found that schools reduced the percent of products that were high-in from 90.4% before implementation to 15.0% after implementation [70], which may have influenced children's preferences and the subsequent household purchase made by their parents. Indeed, a separate qualitative study found that schools were a key promotor of behavioral change relating to the labeling component of the law: children learned about the regulation at school and then encouraged their mothers to purchase nonlabeled foods and beverages for them [71]. In addition, an evaluation of Chile's food marketing regulations found that children and adolescents' exposure to unhealthy food and beverage advertisements on television was reduced by 44% and 58%, respectively, in the year following implementation of the law [72], although it is unclear how reductions in exposure translate to changes in purchasing behavior. Another consideration is that as these regulations were being enforced, the Chilean Ministry of Health launched a mass media campaign to inform consumers about the meaning of the warning labels and encourage them to choose products with fewer labels, potentially further strengthening the effect of the law. Unfortunately, we are unable to disentangle the effects of each of the labeling, marketing, and school sales ban components of the law, here. Future research will be needed to understand the individual impact of each policy as well as how the policies and media coverage and promotion interacted to lead to reductions in high-in beverage purchases.

This study has important limitations. As previously mentioned, the main limitation is that this is an observational pre-post study and thus unable to assess the causal impact of the law or disentangle the drivers of the observed reductions in high-in beverage purchases. The reductions in purchases of high-in SSBs found in this study likely reflect a combination of changes in consumer behavior (e.g., consumers choosing not to purchase a high-in beverage) and

industry behavior (e.g., product reformulations that could shift products from high-in to not-high-in status, or other industry actions such as changes in marketing strategy or pricing changes). For example, after the policy, we observed that there was a large decline in the percent of households who purchased high-in fruit drinks and a sizeable increase in the percent who purchased high-in fruit drinks after the policy. Currently, it is not clear whether this is because consumers were choosing to switch from high-in fruit drinks to not-high-in fruit drinks or whether they purchased the same or similar beverages, but these beverages were reformulated under the nutrient thresholds and thus no longer subject to the regulation (and thus no longer classified as "high-in"). Future research to understand the mechanisms of how this regulation changed beverage purchases will be important for designing future policies. For example, regulations intended to incentivize reformulation may be quite different than those designed to influence consumer behavior.

It is also important to note how different analytical approaches may have affected results. For example, the difference between observed and expected purchase amount depends on a comparison of the slope of the preregulation trend with the slope of the postregulation trend. Underestimating the preregulation trend could, therefore, lead to an underestimated counterfactual and, ultimately, an overestimation of the difference between slopes in the postregulation period.

Similarly, the specified length of the preregulation period can affect the preregulation trend, thereby affecting the counterfactual comparison. One potential limitation of this study is that we have a relatively short preregulation time period (January 2015 to June 2016). Notably, results were consistent even in sensitivity analyses in which we expanded the preregulation period to January 2014 to account for Chile's October 2014 SSB tax modification (i.e., we account for the fact that there may have been a steeper preregulation trend between October 2014 and June 2016 due to the change in SSB tax). It will be important for future policy research to consider the effects of including different preregulation time periods on results, especially in countries where multiple policies are implemented over time.

Our analyses also included only a year and a half of postimplementation data. The results show a significant reduction in purchases of high-in beverages immediately after the regulation was implemented, with no change in the postregulation trend (i.e., there was no attenuation or increase in effect over time). These results could be due to the relatively short postimplementation period. Over time, we might expect to see an increase in the effect as social norms shift and as people learn more about the health harms of high-in beverages. Alternately, we could see the effect decrease if consumers grow accustomed to the warning labels and the "novelty effect" wears off, as has been shown in tobacco labeling [73]. Moreover, the nutrient thresholds became stricter (i.e., more products became high-in and were subject to the warning label and marketing and school restrictions) in June of 2018. Longer-term analyses will be needed to understand the effects of the law as it gets stricter over time, as well as to understand whether purchases of high-in beverages remain low, continue to decline, or rebound over time.

Another limitation is that our work evaluates changes in high-in beverages without regard for which specific nutrient(s) exceeded thresholds (sugar, saturated fat, sodium, and/or calories). Although for beverages, the vast majority were regulated as high-in because of excess sugar content, future research should explore potential differences in purchasing according to specific high-in nutrient or the number of high-in nutrients, since the more nutrients that are regulated, the more warning labels a product will carry.

The study also has several limitations relating to generalizability. For example, the sample is limited to only urban-dwelling households. However, 90% of Chile's population is urban, suggesting that the results are generalizable to most Chilean households [41]. Regardless, it will be important for future research to understand whether policies affect purchases differently in rural areas versus urban cities, as well as by geographical region. Similarly, the study only includes beverage

purchases made at stores, including supermarkets, grocery stores, and convenience stores. Store-bought beverages account for roughly 88% of nonalcoholic beverage sales in Chile, and this proportion has remained consistent before and after the regulation [74]. However, future research should examine whether there have been any changes in the type or quantity of beverages purchased from other sources (e.g., in restaurants, schools, or homemade), as well as whether actual dietary quality of beverages across sources has changed. For example, if the healthfulness of store-bought beverages improved, but the healthfulness of restaurant beverages declined, this could offset improvements in diet quality and blunt any subsequent health gains. Additional work will also be needed to understand changes in food purchases and food intake after these regulations, as well as how consumers potentially substituted between foods and beverages. Examining the effects of these regulations across total purchases and diet will be essential for understanding whether this package of policies is likely to achieve its goal of obesity prevention.

It is also important to note that this study evaluated beverage purchases only after the first phase of implementation in 2016. In July 2018, the second phase was implemented, including stricter nutrient thresholds as well as a new marketing law that restricts from 6:00 AM to 10:00 PM all television and cinema advertising for high-in foods and beverages and requires health promotion messages to be included when products are advertised outside those hours. This 6:00 AM–10:00 PM restriction is considered to be the most comprehensive marketing restriction on unhealthy foods and beverages implemented to date [28]. In July 2019, the third and final phase of the regulation was implemented with the most stringent nutrient thresholds. Future research on both foods and beverages will be needed examine the impact of the Chilean regulation following implementation of the new marketing law and the final set of nutritional thresholds in 2019 [31] and whether the effects of these regulations wear off or accelerate over time.

This evaluation has important policy implications. Three other countries (Israel, Uruguay, and Peru) [75] have adopted FOP mandatory warning labels, and Mexico and Brazil are among countries considering warning labels and are in consultative stages. However, the majority of these adopted or proposed regulations focus on warning labels only and do not include other components of the Chilean regulation, such as the restrictions on marketing or school sales and promotions. Multicountry evaluations may be helpful to disentangle the effects of each of these regulations, as well as to understand how the regulations work in different populations.

In conclusion, this study describes changes in SSB purchases following introduction of Chile's policy package that includes FOP warning labels, child-directed marketing restrictions, and restrictions on sales in schools of unhealthy foods and beverages. After implementation of this policy package, purchases of high-in beverages declined by nearly 24%; these reductions are larger than those observed after standalone SSB reduction policies in Latin America, such as taxes. Future research should examine the differential effects of the labeling, marketing, and school policies; whether changes in purchases were driven by industry or consumer behavior; and the effects of these policies on SSB intake.

## Supporting information

**S1 Table. Nutrient thresholds and implementation dates of the Chilean labeling and advertising law.**
(DOCX)

**S2 Table. Beverage groupings.**
(DOCX)

**S3 Table. Twelve-month and 18-month analysis.**
(DOCX)

**S4 Table. Unadjusted percent of consumers who purchased high-in and not-high-in beverages, overall and by beverage type, pre- and postregulation.**
(DOCX)

**S5 Table. Unadjusted mean beverage purchases pre- and postregulation.**
(DOCX)

**S6 Table. Coefficient estimates from the models to estimate changes in purchases of high-in, not-high-in, and total beverages.**
(DOCX)

**S7 Table. Coefficients for the fully interacted model with education level to estimate changes in purchases of high-in beverages.**
(DOCX)

**S8 Table. Model testing.**
(DOCX)

**S9 Table. Sensitivity analysis.**
(DOCX)

**S1 Fig. Chilean front-of-package warning labels.**
(DOCX)

**S2 Fig. Chilean regulation timeline and study data collection periods.**
(DOCX)

**S3 Fig. Monthly unadjusted weighted mean purchase volume of beverages, 2015–2017.**
(DOCX)

**S4 Fig. Mean changes in purchase volume of high-in beverages, stratified by tertile of household assets index.**
(DOCX)

**S1 STROBE checklist.**
(DOC)

## Acknowledgments

We would like to thank Ms. Emily Busey for her contributions proofreading and organizing files and Dr. Donna Miles for data organization and descriptive analyses.

## Author Contributions

**Conceptualization:** Lindsey Smith Taillie, Marcela Reyes, Barry Popkin, Camila Corvalán.

**Data curation:** Lindsey Smith Taillie.

**Formal analysis:** Lindsey Smith Taillie, M. Arantxa Colchero.

**Funding acquisition:** Lindsey Smith Taillie, Barry Popkin, Camila Corvalán.

**Investigation:** Lindsey Smith Taillie, M. Arantxa Colchero.

**Methodology:** Lindsey Smith Taillie, Marcela Reyes, M. Arantxa Colchero, Barry Popkin, Camila Corvalán.

**Project administration:** Lindsey Smith Taillie, Marcela Reyes, Barry Popkin, Camila Corvalán.

**Resources:** Marcela Reyes.

**Supervision:** Marcela Reyes.

**Writing – original draft:** Lindsey Smith Taillie.

**Writing – review & editing:** Lindsey Smith Taillie, Marcela Reyes, M. Arantxa Colchero, Barry Popkin, Camila Corvalán.

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
