## [Decision Letter · Decision Letter 0]

26 Sep 2019

Dear Dr. Taillie,

Thank you very much for submitting your manuscript "An evaluation of Chile’s front-of-package warning label policy on sugar-sweetened beverage purchases: a pre-post study" (PMEDICINE-D-19-01988) for consideration at PLOS Medicine. 

[LINK]

In light of these reviews, I am afraid that we will not be able to accept the manuscript for publication in the journal in its current form, but we would like to consider a revised version that addresses the reviewers' and editors' comments. Obviously we cannot make any decision about publication until we have seen the revised manuscript and your response, and we plan to seek re-review by one or more of the reviewers. 

We expect to receive your revised manuscript by Oct 10 2019 11:59PM. Please email us (plosmedicine@plos.org) if you have any questions or concerns.

We look forward to receiving your revised manuscript. 

Sincerely,

Adya Misra, PhD

Senior Editor 

PLOS Medicine

plosmedicine.org

-Title- please consider including " before-and -after study" 

- Please consider combining first two sentences of the Abstract Methods and Findings.

-Please provide basic demographic details in the abstract

- Please provide p values with 95%CIs where appropriate

- Please provide a completed STROBE checklist to comply with our reporting requirements. Please ensure that you don't use page numbers as these are likely to change 

- as noted by reviewers, a reference listed as under review. Please note this is not permitted as per PLOS Data policy. We would recommend providing further details regarding this reference 

-Please state "study limitations include...." or something similar around line 411

Comments from the reviewers:

Reviewer #1: The stated purpose of this manuscript is to evaluate the impact of Chile's labeling and advertising law on household beverage purchases. One major concern is that the policy(s) of interest change throughout the manuscript and co-occurring policies/practices in Chile make it impossible to tease out the impact of the front-of-package warning label on household purchases of SSBs. With respect to the changing policy of focus for the evaluation, the paper title and abstract point only to the association between the Chilean front-of-pack labeling policy and household SSB purchases while the intro and discussion point to the association between both the Chilean labeling and marketing regulations and household SSB purchases. With respect to co-occurring policies/practices, the introduction also mentions a third policy in Chile - restrictions on the promotion and sales of certain foods and beverages in schools which went into effect at the same time as the marketing regulation (June 2016). Based on the study they cite, it is probable that this school policy may also have effected household SSB purchases. In addition, the discussion mentions a mass media campaign by the Ministry of Heath to inform customers about the meaning of the warning labels and encouraging them to choose products with fewer labels. The combination of the observational study design and co-occurring policies/practices makes it very hard (if not impossible) to interpret the observed decline in SSB purchases. And the fact that the authors observe a much bigger impact than a recent meta analyses on food labeling (although it was not just FOP) suggests this is not a pure FOP effect (which the authors acknowledge may be the case). 

To address the two concerns described above, the paper should be recast to clearly focus on all the co-occurring policies/practices in Chile, not just FOP. And since the UNC team has considerable data from various countries in South America, it would be ideal to include data from a comparable control country without the same suite of policies as Chile and conduct a natural experiment with a difference-in-differences approach. This would be much more robust analytical approach, and while it could not tease apart the impact of the various policies/practices on household SSB purchases, it could answer the question of whether the observed changes in household SSB purchases in Chile are due to the combination of policies/practices or just secular trends in the region.

Another major concern is the time frame of the pre and post data. The pre period is from January 2015 to June of 2016 and the post period is from July 2016 to December 2017. While the authors do adjust for month to account for seasonality, the pre and post periods are not overlapping in terms of calendar months. This is a big issue since lots of literature has shown that people purchase beverages differently depending on the time of year. And beverage purchases may be particularly sensitive to weather, month, holidays etc. Ideally, the authors should have one full year prior and one full year post. If that is not possible, the authors should use the same months for the post period (January 2016 to June of 2017). Without making this change, it is not possible to interpret the observed decline.

A more minor point is that since their Kantar panel only includes store bought beverages and not purchases outside the home such as at restaurants or street vendors, it would be helpful to add a sentence to the introduction about what fraction of beverage sales are store bought. If it is small, that should be included as a limitation. 

Another minor point is that the authors cite a paper under review. I would remove this cite and the associated evidence until it is in press. 

Reviewer #2: This is a good attempt to evaluate an complex, but internationally important, public health intervention. Further justification of the approach taken and clearer conceptualisation of the intervention and its potential impacts would be valuable.

Substantive conceptual comments

The conceptualisation of exactly what the intervention of interest is is inconsistent. Only warning labels are mentioned in the title and abstract. Concurrent marketing restrictions are mentioned in the introduction. A concurrent mass media education campaign is mentioned in the discussion. In some places the results are attributed simply to the warning labels, in other places the marketing restrictions are also recognised as playing some role, in one place it's acknowledged that it's impossible to "differentiate effects" of the intervention components. I think it would be more honest to provide a much clearer and consistent conceptualisation of the intervention as the combination of the simultaneous introduction of warning labels and marketing restrictions alongside mass media education. This will require some re-pitching of the introduction; refinement of the text throughout; and further consideration of what the relevant comparator literature is in the discussion. It is not surprising to me that a multi-component strategy introduced in the context of a pre-existing SSB tax has a greater effect than a tax alone - and the need for multi-component strategies, rather than single 'magic bullet' interventions, is an incredibly important policy message.

The rationale for focusing on beverage purchasing specifically, rather than purchasing of all labelled food, is not clear to me. Whilst SSB consumption has clearly been a particular area of recent policy focus, I'm not sure we should uncritically privilege it as the main public health problem of our time. Perhaps some evidence to indicate the PAF of eg obesity or diabetes for SSBs might provide some justification? Do you have comparable data for food that could be included? If so, I'd prefer to see that here so we can get a more comprehensive picture. 

An implicit rationale for focusing on beverage consumption appears to be the ability to contrast effects of taxes against the current suite of interventions (as per Fig 3). I think this is an unfair comparison in this context. The current interventions were introduced in the context of an existing tax - this is not an either/or situation. It is entirely feasible that the interventions studied here would not have had the same effect without the tax also being in place. 

The exclusion of out-of-home purchases seems a major limitation that little attention is paid to. Are there other sources of data that you could use to study this? Can you say a little more in the discussion about what biases this might impose? 

I'm not sure why average purchases across the whole post intervention period vs counterfactual is the most appropriate way of expressing impact. I would have thought this mean is heavily influence by the length of time for which post-intervention data available. I wonder if a more appropriate approach would be to choose a policy relevant 'endpoint' for the study and estimate change between observed and counterfactual at that point. Perhaps immediately prior to implementation of next regulations would be sensible - which is the most sustained impact of the current intervention that can be measured before an additional intervention is added.

Other comments

The manuscript includes numerous typos, repeated words, missing 'minus' signs, trailing phrases, and even one "ref" where a reference seems yet to be added. 

I'm not sure that "fiscal policies have been the major approach for reducing [SSB] intake". They have been a big focus recently. But prior to that there were e.g. educational campaigns related to dental health and sugary drink consumption; and even now I think governments might say they are doing quite a lot to reduce consumption that isn't taxes - education, labelling, supporting reformulation, encouraging reduce package size etc.

I'm also not sure that "front of package warning labels, marketing restrictions, and bans" have been "more recently" discussed. I think these have been considered at least as long as fiscal policies. 

Whilst I agree that "education may influence purchasing decisions by affecting how well an individual is able to understand the information communicated in nutrition messages", I think a major additional issue here is that education is likely to be associated with greater (financial and non-financial) resources that allow individuals to enact the changes encouraged. 

If I understand supplemental fig 2 correctly, the guidance on the intervention assessed here was published in June 2015 - six months after the pre-data phase began. It seems possible that this publication represented a further 'intervention' with the potential to change both industry and consumer behaviour. From the authors knowledge of the context, is this feasible? If so, how does the presence of this additional intervention in the pre-period impact interpretation?

I don't understand why "a team of Spanish-speaking nutritionist research assistants" were needed to code products as label-able or not. Wasn't this an automated calculation done using NFP data?

It would be helpful to label the components of the analysis (descriptive/unadjusted, fixed effects/adjusted) similarly in the methods and results to allow readers to easily join the two up.

If I understand correctly, your data is per household per week, yet the results refer to per capita per day. It wold be helpful to know exactly how you get from one to another and any limitations of this (e.g. is data on household size accurate and is it fair to assume equal distribution of purchases).

The differential results by education appear to be descriptive only. I notice that confidence intervals don't overlap, but wondered if a formal test for interaction would be valuable.

Two evaluations of the impact of the Chile SSB tax were published simultaneously. However, only one is referred to in the comparison of effects.

I think the asterisks in supplemental table 5 title and table body may refer to different things, but the footnotes don't make this clear. Also not clear what is presented in square brackets in this table. Please check through all tables for similar problems.

In addition to the unadjusted trend plots in supplemental fig 3, I would like to see conventional ITS figures plotting adjusted pre and post trends alongside post counterfactual. This would allow readers to helpfully visualise eg acceleration of pre-existing trends.

I think it is incorrect to conclude that "similar absolute changes by household education suggests that this regulation is not likely to increase disparities". If inequalities are measured on a relative scale then similar absolute reductions can result in increased relative differences.

It's mentioned in the discussion that you cannot disentangle consumer vs industry changes. However, I think from the NFP data you have, you could provide some insight into the extent of reformulation. Could that be included here?

I don't see how the interpretation that the "policies accelerated pre-existing trends" is "another interpretation". Surely the only interpretation is there were pre-existing trends, they were steeper post intervention. 

Reviewer #3: This is an important study that evaluates the effect of a multitude of labeling and marketing policies on beverage purchases in Chile. These policies are unique and were the first of their kind to be implemented. The dataset used seems comprehensive and adequate for the this evaluation, and the methods used are the ideal for this type of analysis. Authors explored an extensive range of sensitivity analyses to determine if results were robust to different model specifications and data manipulations, and they preregistered their analytic plan. Weaknesses are clearly articulated and common with this type of observational research. Few suggestions and questions:

1) The authors cite a publication that further describes the database used for their evaluation. It might be helpful to get some basic information on that source into this manuscript (understanding word limitations are a challenge). For example, it would be helpful to know the survey response rate and whether there were specific restrictions/exclusions for data collection.

2) Was any information on policy compliance available from the data collectors who evaluated products in homes? I would imagine that some of the products that met warning label thresholds weren't labeled, and these products might have systematically differed from those that were labeled. This should have biased results to the null but would be interesting to see nonetheless, if it was available.

3) In an interrupted time series analysis, investigators examine the post-intervention level change as well as the trend change (effect size from the indicator of pre v. post as well as the interaction term with time). Those results seem to be in Supp Table 5, and if my interpretation is correct, there is a pre-post level change but no real trend difference pre vs. post. Is that correct? Could be highlighted in manuscript, esp bc suggests no clear attenuation of effect over time (though still only a short period of post-implementation data). 

4) Authors display in Table 1 that the survey responders were different pre vs. post. Any sense of why this might have happened? Stratifying the results by education and assets, as the authors do, certainly helps this, but it does seem peculiar to have such an increase in the education level post-regulation. Was there a requirement, for this study, that panel members had some data in the pre and post period, or was all data included regardless of whether participants were in both periods?

5) This is really a style preference, but some of the top line results presented are relative instead of absolute changes. I think the absolute changes might be more helpful (might just require switching out the figures and presentation of result to show the absolute differences as the main outcome). The comparison with other policy effects is very useful and puts the small changes here into context - small but important at the population level and larger than what was seen from the SSB tax in Mexico.

6) The differences reported are at the population level. This obviously includes some households that had no purchasing of labeled beverages. Would be helpful to also see what the changes were in households that had pre-implementation purchases of labeled beverages.. 

7) Supplemental Figure 3 shows an uncharacteristic spike in ml purchased in Jan 2017. Any thoughts about what that represents? Could be a New Year's effect but don't really see in prior years, during the pre-reg period. 

8) The absolute changes in % of consumers of fruit drinks and dairy pre vs. post are stunning. Could be highlighted in the manuscript more.

Jason Block

Reviewer #4: See attachment

Michael Dewey

[LINK]

---

## [Decision Letter · Decision Letter 1]

26 Nov 2019

Dear Dr. Taillie,

Thank you very much for re-submitting your manuscript "An evaluation of Chile’s Law of Food Labeling and Advertising on sugar-sweetened beverage purchases: a before and after study" (PMEDICINE-D-19-01988R1) for review by PLOS Medicine.

I have discussed the paper with my colleagues and the academic editor and it was also seen again by 4 reviewers. I am pleased to say that provided the remaining editorial and production issues are dealt with we are planning to accept the paper for publication in the journal. 

[LINK]

We look forward to receiving the revised manuscript by Dec 03 2019 11:59PM. 

Sincerely,

Adya Misra, PhD

Senior Editor 

PLOS Medicine

plosmedicine.org

Requests from Editors:

Title- please add dates of study to the title

We appreciate your position on p-values and how they are often misconstrued to mean scientific significance rather than statistical. However, to allow comparison between studies and in order for the broad readership of PLOS Medicine to understand your findings, we ask that you provide p-values along with 95% confidence intervals. 

DAS- please clarify if an accession number is required when requesting the study data

Please include full stop after the square brackets

Line 321-322 contains a reference to “results not shown”. Please note that in order to adhere to PLOS data policy, we require all the underlying data in the article to be provided within the main text or SI files. 

Discussion needs to be toned down, especially the sentence “household purchases of high-in beverages decreased 23.7% beyond what would have been expected had the regulation not been enacted”. Please include details to provide context such “according to our counterfactual model…” or similar 

Line 470 onwards needs to be toned down, as a direct cause and effect cannot be determined from this study design

Line 585- please avoid assertions of primacy 

Comments from Reviewers:

Reviewer #1: The authors have done a good job of responding to the reviewer comments. The one area where I am not convinced is related to the time frame of the pre/post periods. And I think this is a significant issue. This was was raised by reviewer #1. The authors use 18 months before and 18 months after which means that the pre and post periods are non-overlapping in terms of calendar months. SSB purchases are very seasonal, so I agree that it would be best to use the exact same time frame before and after. Identical months in the pre and post period is critical for accounting for some of the big limitations that are inherent in natural experiments. The authors say that they ran their model with different window time periods to test the robustness of the results. And they say that when they used the 12m time frame, the results jump around. I am guessing the positive results when they use the month dummies is in large part due to the spike in Jan 2017 (which Reviewer #3 pointed out and it sounds like the Jan spike is actually a February spike due to miss-coding). I wonder if it might be worth digging into the data miss-classification issue in January/February to see if there are other issues there. At the very least, the authors need to present the sensitivity analyses using 12m rather than just describe the results in words without specific point estimates. In terms of the best model for the main analysis, I would suggest using 12m before and 12m after with the month dummies. 

Reviewer #2: Thanks for responding to my previous comments. I don't have any further comments.

Reviewer #3: Jason Block

Really important work! Manuscript is improved with the recommended changes, and authors have been very responsive. A few small suggestions: 

Page 32, line 88 - I think there might be different descriptions of sugar content to be classified as "high in" - is it 22.5 or 6g? Maybe there is some switching of the requirements for food and liquids per Table S1?

Page 37, line 298 - There is some presentation of the decline in consumers of high in beverages, but this seems to only be available as unadjusted data? Why not show for the adjusted results as well. I might also present in the text the % of consumers of high in beverages before and after (you show change only), like you do for the not high in.

In Table 1, it might be helpful to present the mean consumption per capita during the pre period, as you do in S6 and S7 - this is really useful info. I only say this bc your relative change post regulation is large, but the absolute change in ml is small. Might allow for better comparison with other countries, such as US, where consumption is very high. 

As for S6, can the decline in fruit drink consumers be real? I know fruit drink consumption is low overall, but the % of consumers is still reasonable high, and the drop off is dramatic. Might be worth at least some comment in discussion.

Page 41, Figure 3 - Might be worth a mention somewhere that the Chilean and Mexico taxes were pretty small and that larger taxes (Philly, for example) are showing large relative changes in consumption, as would be expected. You mention that the Chilean tax was small but not the Mexico tax. Could even be a footnote in this figure, to put in context of other taxes, or you could mention in the discussion.

Reviewer #4: The authors have addressed my detailed comments.

One issue remains. I suggested that since the results were surprising in the extent of the reduction found for SSB that light might be thrown on this by examining the effect of the labels other than high in sugar. I appreciate this would mean looking at other food groups than SSB. I did not find the authors' rebuttal too convincing on this point but I feel this is not purely a statistical question so I am happy for the editorial team to decide.

For what it is worth I sympathise with the authors' reluctance to add p-values everywhere in addition to the confidecne intervals.

Michael Dewey

[LINK]

---

## [Editor Report · Decision Letter 2]

7 Jan 2020

Dear Dr. Taillie, 

On behalf of my colleagues and the academic editor, Dr. Sanjay Basu, I am delighted to inform you that your manuscript entitled "An evaluation of Chile’s Law of Food Labeling and Advertising on sugar-sweetened beverage purchases from 2015 to 2017: A before and after study" (PMEDICINE-D-19-01988R2) has been accepted for publication in PLOS Medicine. 

PRODUCTION PROCESS

PRESS

PROFILE INFORMATION

Thank you again for submitting the manuscript to PLOS Medicine. We look forward to publishing it. 

Best wishes, 

Adya Misra, PhD

Senior Editor 

PLOS Medicine

plosmedicine.org